# Hypolipidemic, Antioxidant and Immunomodulatory Effects of *Lactobacillus casei* ATCC 7469-Fermented Wheat Bran and *Spirulina maxima* in Rats Fed a High-Fat Diet

**Asmaa Abdella [1,*], Mohamed F. Elbadawy [2], Sibel Irmak [3] and Eman S. Alamri [4]**

[1] Department of Industrial Biotechnology, Genetic Engineering and Biotechnology Research Institute, University of Sadat City (USC), Sadat City 32897, Egypt

[2] Department of Microbiology and Immunology, Faculty of Pharmacy, University of Sadat City (USC), Sadat City 32958, Egypt

[3] Department of Agricultural and Biological Engineering, Pennsylvania State University, State College, PA 16802, USA

[4] Nutrition and Food Science Department, University of Tabuk, Tabuk 47512, Saudi Arabia

[*] Correspondence: asmaa.abdelaal@gebri.usc.edu.eg

**Abstract:** Hyperlipidemia is a leading cause of atherosclerosis and coronary heart disease (CHD). This study aimed to investigate the hypolipidemic effect of *Lactobacillus casei* ATCC 7469-fermented wheat bran extract and *Spirulina maxima* extract on Sprague–Dawley rats fed a regular or high-fat diet compared to rosuvastatin as a reference drug. Treatment with *Lactobacillus casei* ATCC 7469-fermented wheat bran and Spirulina maxima resulted in a significant decrease in total cholesterol (TC), triglycerides (TG.), low-density lipoprotein (LDL) and very low-density lipoprotein (VLDL) ($p < 0.05$) and a significant increase in high-density lipoprotein (HDL) ($p < 0.05$). That combination also improved liver functions. It also resulted in the improvement of liver oxidative biomarkers and decreased the production of inflammatory markers (TNF-$\alpha$, IFN-$\gamma$, IL-10, and IL-1$\beta$). In addition, a significant reduction in inflammation of liver tissues was observed after that treatment. *Lactobacillus casei* ATCC 7469-fermented wheat bran extract and Spirulina maxima extract had additive effects on the lipid profile, liver functions and immune system of rats similar to rosuvastatin.

**Keywords:** wheat bran; fermentation; *Lactobacillus casei*; *Spirulina maxima*; lipid profile; liver function; antioxidant; immunomodulatory





## 1. Introduction

Hyperlipidemia is the most common cause of atherosclerosis, which is connected to coronary, cerebrovascular and peripheral vascular illnesses in ischemic heart disease (IHD) [1]. Additionally, it leads to oxidative stress, which generates a significant number of oxygen free radicals and causes (LDL-C) to undergo oxidative alterations [2]. To reduce the risk of CHD, patients with hypercholesterolemia must have lower blood cholesterol levels. Numerous medications, including statins [3], bile acid sequestrants [4], fibrates [5] and nicotinic acid [6], are available to treat hypercholesterolemia. However, they can cause adverse side effects, including myalgia, a lack of strength, headaches and hyperglycemia [7].

Wheat bran (WB) has attracted much interest since it is the most significant milling byproduct of grain cereals. Wheat bran is a complex substrate made mostly of dietary fiber proteins and carbohydrates, as well as vitamins, minerals and phenolic acid compounds, primarily ferulic acid and p-coumaric acid [8]. Microbial fermentation can produce medicinally high-value products [9]. Probiotics such as Lactobacillus are living microbial feed supplements that have a beneficial effect on the host animal's intestinal balance [10]. Several studies have demonstrated that the fermentation of cereals by Lactobacillus results in the production of several bioactive compounds. The fermentation products of rice bran

by Lactobacillus were found to reduce body weight and improve lipid metabolism [11]. Luana et al. [12] found that oat extract fermented by Lactobacillus had a 40% higher content of soluble b-glucan, which reduced the cholesterol level in hyperlipidemic patients more than unfermented extract. Shahbazi et al. [13] reported that probiotic fermentation of food enhances its anti-inflammatory and immunomodulatory properties.

Cyanobacteria, or blue-green algae, are a class of Gram-negative photoautotrophic prokaryotes that have a blue-green pigment (c-phycocyanin) [14]. Spirulina is a blue-green microalga that is rich in antioxidants, proteins, minerals, fatty acids and vitamins [15]. Recent studies have revealed that spirulina can enhance the nutritional qualities, probiotic viability and functional product features of fermented milk [16]. Spirulina has been shown to have antimicrobial, anti-inflammatory, anticancer, hypolipidemic and hypoglycemic properties [17]. Abdel-Daim et al. [18] stated that Spirulina platensis exhibited significant modulatory and anti-inflammatory effects on acetic acid-induced colitis in rats.

To the best of our knowledge, this is the first study to evaluate the synergic hypolipidemic, antioxidant and immunomodulatory effects of a combination of *Lactobacillus casei* ATCC 7469-fermented wheat bran extract and *Spirulina maxima* extract on rats with diet-induced hyperlipidemia. We also aimed to investigate their synergic effect on liver functions in a rat model.

## 2. Materials and Methods

### 2.1. Bacterial Strains

*Lactobacillus casei* ATCC 7469 was obtained from the American Type Culture Collection, (Manassas, VA, USA). The cells were cultured in MRS broth at 37 °C for 24 h. After cultivation, the cells were harvested by centrifugation at $3000 \times g$ for 20 min and washed with sterile distilled water. The cells were lyophilized and stored at $-80$ °C.

### 2.2. Preparation of the Wheat Bran Fermentation Extracts

Glycerol-preserved *Lactobacillus casei* ATCC 7469 was cultured in MRS medium at 37 °C for 8 h. 2% ($V/V$) *Lactobacillus casei* was inoculated into the fermentation medium (5 g of wheat bran, peptone 10, $NaH_2PO_4 \cdot 2H2O$ 2.0, $MgSO_4$ $7H_2O$ 0.1, $MnSO_4 \cdot 4H_2O$ 0.05 and 50 mL of distilled water) at Ph = 6.5 and incubated at 37 °C for 24 h. Finally, the supernatant was collected by centrifugation (12,000 rpm, 30 min, 4 °C). The extract was vacuum-dried until a constant weight was reached.

### 2.3. Algae Cultivation

*Spirulina maxima* cyanobacteria were cultivated in Zarrouk's medium [19] at 25 °C $\pm$ 2, pH 10, with continuous illumination using cool white fluorescent tubes (2500 Lux) and shaken by hand twice daily for 15 days. Cells were collected by centrifugation (10,000 rpm, 20 min, 4 °C) and freeze-dried into powder before use.

### 2.4. Experimental Design for Rats Fed Lactobacillus casei ATCC 7469-Fermented Wheat Bran Extract and Spirulina maxima Extract

#### 2.4.1. Animals

Thirty male Sprague–Dawley rats, weighing approximately 80–90 g at the age of 8 weeks, were obtained from the Faculty of Veterinary Medicine, University of Sadat City, Sadat City, Egypt. All animal handling procedures, sample collection and disposal were performed according to the regulation of the Institutional Animal Care and Use Committee (IACUC), Faculty of Veterinary Medicine, University of Sadat City, Egypt, under approval number VUSC-019-1-22. All experiments were performed in accordance with relevant guidelines and regulations.

The rats were randomly housed in polypropylene cages at a temperature of 25 °C $\pm$ 2 and a light period of 12:00 to 12:00 and permitted to acclimatize for 10 days before the experiment. During the 10-day acclimation period, the animals were fed diet ad libitum

daily. Then, the rats were randomized into six groups (*n* = 5/group) and treated daily for seven weeks as follows.

**G1**: Normal group: (Fed regular diet).

**G2**: Positive control: (Fed a high-fat diet)

**G3**: Fed a high-fat diet with (10 mg/kg B.W.) of *Spirulina maxima extract*

**G4**: Fed a high-fat diet with 1.0 g/kg body weight of *Lactobacillus casei* ATCC 7469-fermented wheat bran extract

**G5**: Fed a high-fat diet with (10 mg/kg B.W.) of *Spirulina maxima* extract and 1.0 g/kg body weight of *Lactobacillus casei* ATCC 7469-fermented wheat bran extract

**G6**: Fed a high-fat diet with (10 mg/kg B.W.) rosuvastatin.

The regular diet consisted of wheat flour 22.5%, soybean powder 25%, essential fatty acids 0.6%, vitamins (A 0.6 mg/kg of diet, D 1000 IU/kg of diet, E 35 mg/kg of diet, niacin 20 mg/kg of diet, pantothenic acid 8 mg/kg, riboflavin 0.8 mg/1000 kcal of diet, thiamin 4 mg/kg of diet, B6 50 µg/kg of diet and B12 7 mg/kg of diet) and minerals (calcium 5 g/kg of diet, phosphorus 4 g/kg of diet, fluoride 1 mg/kg of diet, iodine 0.15 mg/kg of diet, chloride 5 mg/kg of diet, iron 35 mg/kg of diet, copper 5 mg/kg of diet, magnesium 800 mg/kg of diet, potassium 35 mg/kg of diet, manganese 50 mg/kg of diet and sulfur 3 mg/kg of diet). The nutritional contents of the high-fat diet were similar to those of the regular diet except for the addition of 177.5 g/kg of lard and 12 g/kg of cholesterol [20].

### 2.4.2. Termination of the Experiment

At the end of the experimental period (7 weeks), rats were fasted overnight, weighed and sacrificed by decapitation under anesthesia using xylazine HCl (10 mg/kg/BW) and ketamine HCl (50 mg/kg/BW).

### 2.4.3. Blood Sampling and Biochemical Parameters

Blood was collected using sodium fluoride as an anticoagulant and centrifuged at 3000 rpm for 30 min. The sera were quickly removed and kept at 20 °C until used for biochemical investigations. The serum triglyceride (TG) concentration was determined according to the method of Fossati and Prencipe [21]. Serum total cholesterol (TC) concentration was determined according to the method of Deeg and Ziegenohrm [22]. Serum HDL concentration was measured according to the method of Burstein et al. [23]. Serum LDL concentration was determined according to Friedewald et al. [24]. Serum very low-density lipoprotein (VLDL) was determined according to Norbert [25]. T.G., cholesterol, LDL, VLDL and HDL assay kits were purchased from (Asan and Youngdong Pharmaceutical Co., Seoul, Korea).

The liver was blotted, weighed and homogenized with phosphate-buffered saline to estimate aspartate aminotransferase (AST) and alanine transaminase (ALT) according to the method of Reitman and Frankel [26] and alkaline phosphatase (ALP) according to the method described by Belfield and Goldberg [27]. AST, ALT and ALP assay kits were purchased from (Abcam, Waltham, Boston, MA, USA). The activity of hepatic superoxide dismutase (SOD) was measured according to the method of Marklund and Marklund [28]. Liver catalase (CAT) was determined according to the technique of Cohen et al. [29]. SOD and CAT assay kits were purchased from Themo Fisher Scientific, (USA). Levels of the proinflammatory cytokines TNF-$\alpha$, IL-1$\beta$, and IL-6 in serum were determined by using commercially available ELISA kits (Abcam, Waltham, Boston, MA, USA) according to the manufacturers' recommendations [30].

### 2.4.4. Histopathological Examination

For adequate fixation, liver tissues were cut and kept in 10% formalin. These tissues were prepared and embedded in paraffin wax. Sections with a thickness of 5–6 microns were cut and stained with hematoxylin and eosin. All tissue slices were evaluated under the microscope using the Bancroft technique [31].

*2.5. Statistical Analyses*

The data were analyzed using one-way analysis of variance (ANOVA) version IA (C). PC-STAT, Program coded by University of Georgia, USA [32].

## 3. Results and Discussion

*3.1. Effect on Lipid Profile*

Rats fed the high-fat diet (G2) had significantly greater T.C., T.G., LDL and VLDL and lower HDL than other groups ($p < 0.05$). Treatments of hyperlipidemic rats with s*pirulina maxima* extract (G3), *Lactobacillus casei* ATCC 7469-fermented wheat bran extract (G4) and their combination (G5) significantly decreased the levels of T.C., T.G., LDL and VLDL in hyperlipidemic rats (G2), while they were still significantly higher than those of control rats (G1) or HFD rats treated with standard atorvastatin (G6) ($p < 0.05$). Treatments of hyperlipidemic rats with s*pirulina maxima* (G3) extract, dried wheat bran fermentation extract (G4) and their combination (G5) significantly increased the level of HDL of hyperlipidemic rats (G2), while they were still significantly lower than that of control (G1) or on HFD rats treated with standard rosuvastatin (G6) (Table 1).

**Table 1.** Effect of *Spirulina maxima* extract, *Lactobacillus casei* ATCC 7469-fermented wheat bran extract and their combination on lipid profiles of Sprague–Dawley rats at 7 weeks.

| Variable with Units | G1 | G2 | G3 | G4 | G5 | G6 |
|---|---|---|---|---|---|---|
| TC (mg/dL) | 85.2 ± 1.15 [a] | 204 ± 2.32 [c] | 124 ± 1.09 [b] | 130.64 ± 1.1 [b] | 92 ± 1.3 [a] | 88 ± 0.3 [a] |
| TG (mg/dL) | 67.86 ± 1.15 [a] | 158.33 ± 1.26 [e] | 85 ± 1.3 [c] | 105.28 ± 1.32 [d] | 73 ± 1.15 [b] | 69 ± 1.23 [ab] |
| LDL (mg/dL) | 33.6 ± 0.7 [a] | 89.06 ± 2.17 [e] | 64.9 ± 2.8 [c] | 72.4 ± 2.16 [d] | 58 ± 1.1 [c] | 49 ± 1.15 [b] |
| VLDL (mg/dL) | 12.76 ± 0.3 [a] | 28 ± 0.51 [e] | 20.28 ± 0.58 [d] | 18.36 ± 1.14 [c] | 14.4 ± 0.24 [b] | 12.8 ± 0.31 [a] |
| HDL (mg/dL) | 23.6 ± 0.21 [a] | 15.12 ± 0.7 [e] | 17.6 ± 0.16 [d] | 18.4 ± 0.18 [d] | 20.52 ± 0.16 [c] | 22 ± 0.22 [b] |

G1: Regular diet (R.D.); G2: High-fat diet (HFD); G3: HFD+ *Spirulina maxima* extract G4: HFD+ *Lactobacillus casei* ATCC 7469-fermented wheat bran extract; G5: HFD+ combination of *Spirulina maxima* extracts and *Lactobacillus casei* ATCC 7469-fermented wheat bran extract; G6: HFD+ rosuvastatin. The results are expressed as the means ± SD ($n = 5$). Different superscript letters in the same row indicate significant differences between different groups of rats at $p < 0.05$.

Fermentation of wheat bran increased soluble dietary fibers (SDF). Through the action of fat-binding molecules, the inclusion of SDF in the diet alters the digestibility of starch and lipids, lowering blood levels of cholesterol [33]. Fermented wheat bran contains a high level of Lactobacillus, which lowers cholesterol levels in rats fed a high-fat diet. The cholesterol-lowering effect of Lactobacillus was achieved by increasing fecal bile acid excretion [34]. The deconjugation of bile and binding to bile acid in the small intestine has been proposed as a possible explanation for lactic acid bacteria's fecal bile acid excretion [35]. Deconjugation of bile acids in the small intestine may result in higher excretion of them [36]. Lactobacillus increased fecal bile acid excretion and hepatic bile acid synthesis by expressing 7-alpha-hydroxylase (CYP7A1), the key enzyme in bile acid metabolism [37]. Lactobacillus also decreased triglycerides by upregulating ApoAV, PPAR and F.X. expression and by increasing apoA-V levels [38,39]. Extraction of cholesterol from the gut was performed by incorporation of probiotics into the cellular membrane [40]. It has also been proposed that probiotics might convert cholesterol to coprostanol, which is then eliminated in the feces, potentially decreasing cholesterol absorption in the gut [41]. Probiotics may also decrease cholesterol by lowering 3-hydroxy-3-methyl glutaryl-CoA (HMG-CoA) reductase activity, a major regulatory enzyme in cholesterol biosynthesis [42]. Kitawaki et al. [43] reported that soy yogurt fermented by lactic acid bacteria fed to rats upregulated the expression of enoyl CoA isomerase, thus increasing the β-oxidation of fatty acids, which indicated that fermented cereals regulated the synthesis and degradation of lipids at the gene level. Gut microbiota can ferment the soluble dietary fibers (SDF) and starch, thus producing short-chain fatty acids, which might have played a crucial role in the reduction of TC [44]. Junejo et al. [45] stated that the fibrous structure of wheat bran has the ability of binding sterol derivatives and cholesterol to inhibit their absorbance in the body

Han et al. [46] reported that a glycolipid derived from Spirulina called glycolipid H-b2 inhibited pancreatic lipase activity in a dose-dependent manner and reduced postprandial TG levels. This action is thought to be secondary to the activation of the AMP-activated protein kinase signaling pathway, which downregulates the expression of lipid synthesizing genes such as sterol regulatory element-binding transcription factor-1c, 3-hydroxy-3-methyl glutaryl coenzyme A reductase and acetyl-CoA carboxylase, lowering TG levels and inhibiting fatty acid synthesis [47]. According to Dvir et al. [48], Spirulina carbohydrates and dietary fibers lower cholesterol by increasing the size of the bile acid pool and fecal steroid excretion. Nagoka et al. [49] discovered that a new protein, C-phycocyanin, produced from Spirulina contains a substantial quantity of cystine and is responsible for increasing HDL-C, and downregulates cofactors in fat metabolism such as adenine dinucleotide phosphate. It also binds to bile acids in the jejunum, affecting the micellar solubility of cholesterol before suppressing the cholesterol absorption.

## 3.2. Effect on Liver Functions

Rats fed the high-fat diet (G2) had significantly greater AST, ALT and ALP levels than the other groups ($p < 0.05$). Rats fed the high-fat diet supplemented with *Spirulina maxima* extract (G3), *Lactobacillus casei* ATCC 7469-fermented wheat bran extract (G4) and their combination (G5) had significantly lower levels of AST, ALT and ALP than rats fed the HFD (G2) ($p < 0.05$). However, they had significantly higher levels than the rats fed the control diet (G1) and the HFD rats treated with standard rosuvastatin (G6) ($p < 0.05$) (Figure 1).

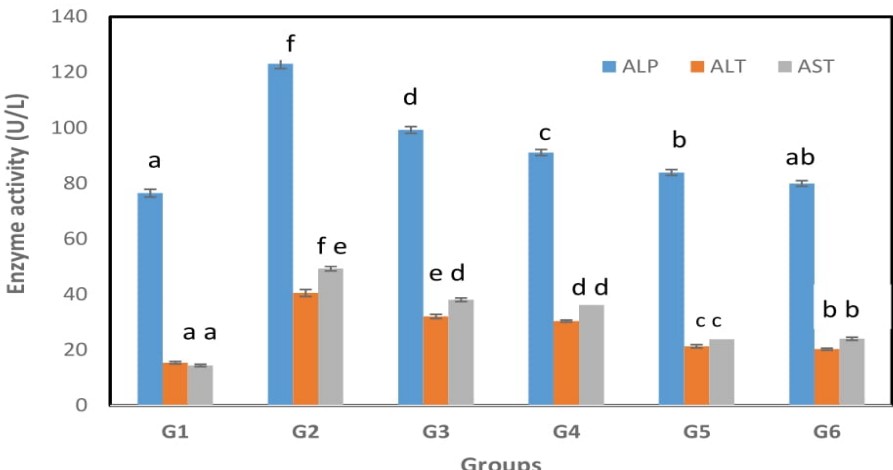

**Figure 1.** Effect of *Spirulina maxima* extract, *Lactobacillus casei* ATCC 7469-fermented wheat bran extract and their combination on liver function enzymes of Sprague–Dawley rats at 7 weeks. G1: Regular diet (R.D.); G2: High-fat diet (HFD); G3: *Spirulina maxima* extract G4: HFD+ *Lactobacillus casei* ATCC 7469-fermented wheat bran extract; G5: HFD+ combination of *Spirulina maxima* extract and *Lactobacillus casei* ATCC 7469-fermented wheat bran extract; G6: HFD+ rosuvastatin. The results are expressed as the means ± SD (*n* = 5). Bars with different superscripts are significantly different (*p* < 0.05).

Hyperlipidemia is correlated with an increase in liver enzymes. The accumulated lipid particles may promote hepatic tissue inflammation by producing free radicals within the liver. These free radicals then cause fibrosis or cell death of the liver tissue. Because of the thinning caused by deposited lipid remnants, the damaged hepatic cell might release more hepatic enzymes outside and become more permeable [50,51]. Furthermore, as cholesterol levels rise, the antioxidant capacity of the human liver may be reduced, resulting in increased inflammation and reactive oxygen species throughout the body, particularly the liver. This might affect various liver activities, including reverse cholesterol transport, and the innate capacity of the liver to maintain LDL levels [52].

*L. plantarum* CQPC03 treatment improved liver function and decreased oxidative stress in mice while also decreasing fat accumulation in the liver. Probiotics substantially

improved liver function parameters [alanine aminotransferase (ALT) and aspartate amino-transferase (AST)] when compared to standard therapy [53,54]. According to research, the mechanisms of liver protection by probiotics mostly include intestinal mucosa repair, lower TNF-levels and increased antioxidant capacity [55]. Chen et al. [56] stated that Lactobacillus could also downregulate the increased levels of AST and ALT after CCL4 treatment. In mice fed a high-fat diet (HFD), Lactobacillus can inhibit liver HMG-CoA reductase and make ferulic acid, which can promote the excretion of acidic sterol, increasing its activity in the treatment of non-Alcoholic Fatty Liver Disease (NAFLD) [57]. The mucosal cells that line the bile system of the liver are the source of ALP, the free flow of bile through the liver and down into the biliary tract and gallbladder that are responsible for maintaining the proper level of this enzyme in the blood. *Lactobacillus acidophilus* was effective in maintaining the integrity and activity of the epithelial cells lining the biliary duct and resulted in decreased levels of ALP [58]. Chen et al. [59] stated that probiotics restore the balance of the intestinal microbiota (symbiotic and pathogenic bacteria), retain the integrity of the intestinal barrier, decrease the production of toxic products and enhance liver function.

Wahida et al. [60] stated that spirulina may scavenge reactive oxygen species generated from it, thereby preventing AST, APT and ALT from leakage into the blood. The hepatoprotective effects of Spirulina are mainly related to C-phycocyanin, β-carotene, vitamin E, gamma-linolenic acid, selenium and chlorophyll, which have antioxidant and anti-inflammatory effects [61,62].

### 3.3. Antioxidant Activity

Rats fed the high-fat diet (G2) had significantly lower CAT and SOD enzymes than other groups ($p < 0.05$). Rats fed the high-fat diet supplemented with Spirulina maxima extract (G3) and Lactobacillus casei ATCC 7469-fermented wheat bran extract (G4) had significantly higher levels of CAT and SOD enzymes than rats fed the HFD ($p < 0.05$). However, they had significantly lower levels than the rats fed the control diet (G1) and the HFD rats treated with standard rosuvastatin (G6) ($p < 0.05$). Rats fed the high-fat diet supplemented with a combination of Spirulina maxima extract and dried wheat bran fermentation extract (G5) had significantly higher levels of CAT and SOD enzymes than rats fed a HFD treated with rosuvastatin drugs ($p < 0.05$) (Figure 2).

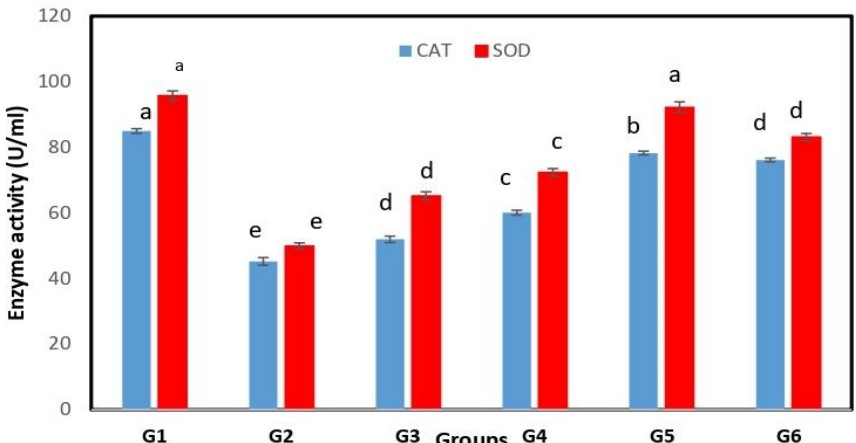

**Figure 2.** Effect of *Spirulina maxima* extract, *Lactobacillus casei* ATCC 7469-fermented wheat bran extract and their combination on liver oxidative enzymes of Sprague–Dawley rats at 7 weeks. G1: Regular diet (R.D.); G2: High-fat diet (HFD); G3: HFD+ *Spirulina maxima* extract G4: HFD+ *Lactobacillus casei* ATCC 7469 fermented wheat bran extract; G5: HFD+ combination of *Spirulina maxima* extracts and *Lactobacillus casei* ATCC 7469 fermented wheat bran extract; G6: HFD+ rosuvastatin. The results are expressed as the means ± SD (*n* = 5). Bars with different superscripts are significantly different ($p < 0.05$).

Protein glycation, glucose-autooxidation and oxidative modification of LDL were all linked to hyperlipidemia, which increased the amount of lipid peroxidation products [63]. Zhang et al. [64] revealed that the use of probiotics significantly ($p < 0.05$) reduces oxidative stress and improves antioxidant capacity. He stated that the antioxidant properties were due to the peroxidation of linoleic acid and the scavenging of superoxide ions and hydroxyl ions. Rehman et al. [65] discovered that Lactobacillus significantly reduced carbohydrate gene enrichment, such as galactose metabolism, which is important in oxidative stress, cognitive impairment and mitochondrial dysfunction. *L. plantarum* NAJU-01 can promote the activity of antioxidant enzymes in rats, regulating the equilibrium of ROS to normal levels in mice. It can also scavenge free radicals and act synergistically with SOD, GSH-Px and CAT to reduce oxidative stress. Wang et al. [66] stated that ROS increased hepatic CYP2E1 protein and mRNA expression and markedly decreased Nrf-2 protein expression, all of which was enhanced by probiotics treatment. Antioxidant activity of Lactobacillus could be attributed to the production of extracellular polysaccharides [67]. The liver protective activity of Lactobacillus acidophilus MTCC447 is by modulation of the antioxidant capacity of the liver and the expression of key apoptotic/antiapoptotic proteins [68].

According to Premkumar et al. [69] *Spirulina fusiformis* increased the activity of cellular antioxidant enzymes such as superoxide dismutase, catalase and glutathione peroxidase to protect against chemically induced hepatotoxicity in rats. Spirulina contains phycocyanin, which scavenges hydroxyl, alkoxyl and peroxyl radicals and prevents lipid peroxidation and iNOS expression in liver microsomes [70]. Another important component of spirulina is β-carotene, which has antioxidant and anti-inflammatory properties and protects against singlet oxygen-mediated lipid peroxidation, making it a useful membrane antioxidant [71]. Flavonoids and phenolic compounds present in Spirulina have good antioxidant potentials because of their hydroxyl groups scavenging ability [72].

*3.4. Immunomodulatory Activity*

Rats fed the high-fat diet (G2) had significantly greater TNF-α, IL-10, IL-6 and IFN-γ levels than the other groups ($p < 0.05$). Treatments of hyperlipidemic rats with *spirulina maxima* extract (G3), *Lactobacillus casei* ATCC 7469-fermented wheat bran extract (G4) and their combination (G5) significantly decreased the levels of TNF-α, IL-10, IL-6 and IFN-γ in hyperlipidemic rats (G2), while they were still significantly higher than those in control rats (G1) or HFD rats treated with standard atorvastatin (G6) ($p < 0.05$) (Table 2).

**Table 2.** Effect of *Spirulina maxima* extract, *Lactobacillus casei* ATCC 7469-fermented wheat bran extract and their combination on inflammation markers in Sprague–Dawley rats at 7 weeks.

| Variable with Units | G1 | G2 | G3 | G4 | G5 | G6 |
|---|---|---|---|---|---|---|
| TNF-α (pg/mL) | 35.4 ± 1.12 [a] | 105 ± 2.4 [f] | 69 ± 2.09 [d] | 72 ± 2.08 [e] | 52 ± 1.56 [c] | 42 ± 1.13 [b] |
| IL-10 (pg/mL) | 5.2 ± 0.2 [a] | 17.68 ± 0.44 [d] | 11.23 ± 0.42 [c] | 13.65 ± 0.4 [d] | 9.2 ± 0.37 [b] | 7.9 ± 0.034 [b] |
| IL-1β (pg/mL) | 3.6 ± 0.1 [a] | 14.9 ± 0.37 [f] | 9.6 ± 0.29 [e] | 8.3 ± 0.22 [d] | 5.1 ± 0.17 [c] | 4.7 ± 0.14 [b] |
| IFN-γ (IU/mL) | 1.53 ± 0.038 [a] | 5.65 ± 0.14 [e] | 3.2 ± 0.08 [d] | 2.9 ± 0.11 [c] | 1.9 ± 0.04 [b] | 1.7 ± 0.05 [a] |

G1: Regular diet (R.D.); G2: High-fat diet (HFD); G3: HFD+*Spirulina maxima* extract; G4: HFD+ *Lactobacillus casei* ATCC 7469-fermented wheat bran extract; G5: HFD+ combination of *Spirulina maxima* extracts and *Lactobacillus casei* ATCC 7469-fermented wheat bran extract; G6: HFD+ rosuvastatin. The results are expressed as the means ± SD ($n = 5$). Different superscript letters in the same row indicate significant differences between different groups of rats at $p < 0.05$.

A number of cells creates cytokines, which are glycoproteins that are released extracellularly to take part in the immune response and control inflammation. IL-1β and TNF-α are frequently recognized as important immunoregulatory cytokines that increase inflammation by triggering a cascade reaction of immune cells and thus promoting the production of cytokines [73]. Hyperlipidemia was proven to initiate an inflammatory response, supported through increased inflammatory biomarkers (TNF-α, IL-1β, IL-6) [74]. Pretreatment with Lactobacillus resulted in downregulation of the expression of tumor necrosis factor-α (TNF-α) [75]. Kitchens et al. [76] found that *Lactobacillus gasseri* SBT2055 was found to

improve the integrity of the intestinal barrier in obese mice, thus inhibiting translocation of lipopolysaccharides (LPS), which form complex with lipoproteins (chylomicrons, HDL, LDL and VLDL), which are transported to liver cells and incorporated by Kupffer cells, leading to increased production of TNF-$\alpha$. The beneficial immune-modulatory effects of Lactobacillus are stimulated through several molecules, such as peptidoglycan and exopolysaccharides, which interact with specific host cell receptors (TLR-2 and TLR-4) [77]. AMPK is an important upstream gene that regulates the balance of lipid metabolism by inhibiting fatty acid and cholesterol synthesis [78]. Activation of AMPK can inhibit the production of TNF-$\alpha$ and IL-6 [79]. Lactobacillus was found to activate the AMPK pathway and result in a reduction in TNF-$\alpha$ and IL-6 [80]. Miyauchi et al. [81] showed that probiotics decreased the secretion levels of TNF-$\alpha$ from the intestinal mucosa and renovated the integrity and barrier function of epithelial cells. Probiotic supplementation decreased TNF-$\alpha$ and TLR (TLR4, TLR5) expression in the intestine and decreased the phosphorylation of p38 MAP kinase in mice with ALD [82]. Lactobacillus-fermented milk contains two soluble proteins, p40 and p75, which have been proved to stimulate survival and growth of intestinal epithelial cells through activation of the epidermal growth factor receptor (EGFR). EGFR and Akt activation prevented cytokine-induced inflammation and intestinal epithelial cell apoptosis [83]

The immunostimulant activity of spirulina is due to its polysaccharide content [84]. There are different mechanisms for enhancing immunity by spirulina, such as the activation of monocytes and macrophages, and increasing interferon production by natural killer cells [85]. The anti-inflammatory activity of Spirulina is probably due to a complex of phycobiliproteins known as the phycobilisome, which consists of erythrophycocyanin, allophycocyanin and C-phycocyanin. The anti-inflammatory activity of C-phycocyanin is due to its reduction in TNF-$\alpha$ levels and myeloperoxidase activity and selective inhibition of key enzymes of acute inflammation, such as COX-2 and iNOS [86]. Chen et al. [87] stated that phycocyanin inhibited expression of genes encoding inducible nitric oxide synthase (iNOS), cyclooxygenase-2 (COX-2), TNF-$\alpha$,IL-6 IL-1$\beta$, IL-2, IL-4, IL-10, IFN-$\gamma$ and IL-17. Phycocyanin also stimulated phosphorylation of inflammation-related signaling molecules such as ERK, JNK, p38 and I$\kappa$B [88].

*3.5. Histopathological Examination of the Liver*

Microscopic examination of the liver in G1 revealed typical hepatic lobules with radiating plates or strands of polygonal cells with conspicuous round nuclei and eosinophilic cytoplasm vertical to the major vein. A discontinuous layer of fenestrated endothelial cells with a delicate arrangement of Kupffer cells lines the sinusoids. The bile duct, portal vein and hepatic artery all had normal histological structures in the portal region (Figure 3a). G2 exhibited hepatic cord disruption as well as necrobiotic alterations in hepatocytes defined by localized necrotic foci and hydropic hepatocyte degradation. Few microvesicular steatosis and apoptotic bodies were observed (Figure 3b). G5 demonstrated marked improvement in the degree of vacuolization and lipid deposition in the cytoplasm of liver tissue caused by the administration of spirulina-and Lactobacillus fermented wheat bran extract (Figure 3c). There was no significant improvement in the histology of G3, G4 and G6 compared to G2. Treatment with *Spirulina maxima*-and Lactobacillus-fermented wheat bran extract may have a good liver protective effect, according to the H&E staining data.

Ramadan et al. [89] stated that the livers of rats fed a high-cholesterol diet exhibited poor cellularity with extensive lipid deposition and enlarged hepatocytes. Lactobacillus was reported to reduce the production of inflammatory factors such as IL-6, IL-l$\beta$, TNF-$\alpha$ and IFN and relieve liver stenosis [90]. Kusumawati et al. [91] found that probiotics improved liver histology, reduced liver fat and reduced liver fibrosis. Yang et al. [92] stated that, after treatment with Lactobacillus of rats fed a high-fat diet, there was no obvious necrosis and slight steatosis, no inflammatory cell infiltration was found, and no fibrous tissue proliferation was found.

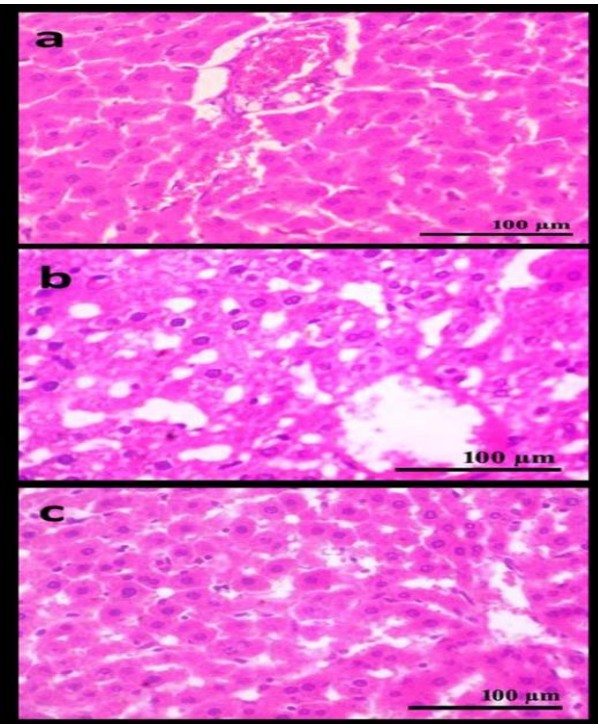

**Figure 3.** Histological sections of rat liver sections stained with hematoxylin and eosin (original magnification ×200) from each group: G1: Regular diet (R.D.); G2: High-fat diet (HFD); G5: HFD+ combination of *Spirulina maxima* extracts and *Lactobacillus casei* ATCC 7469-fermented wheat bran extract (**a**) G1: liver tissue section showed normal histological structure of hepatic lobules (H&E ×200); (**b**) G2: hepatic tissue section showing swelling and vacuolation of hepatocytes with microvesicular steatosis (H&E ×200); (**c**) G5: hepatic tissue section showing mild swelling of hepatocytes with foamy cytoplasm (H&E ×200).

Abdel-Daim et al. [92] stated that by scavenging the free radicals produced during intoxication by various harmful chemicals, spirulina helps to preserve the structural integrity of the hepatocellular membrane. The hepatoprotective effect of spirulina was related to (phycocyanin C), which has antioxidant activity eliminating alkoxyl-, hydroxyl- and peroxyl-free radicals, decreased nitrite production and inhibited hepatic microsomal lipid peroxidation [93].

## 4. Conclusions

Based on the above results, our study may offer novel insights into the potent synergic lipid-lowering, antioxidant and immunomodulatory activities of *Spirulina maxima* extract and Lactobacillus-fermented wheat bran extract supplemented to Sprague–Dawley rats fed a regular or high-fat diet. Administration of these extracts resulted in a significant decrease in total cholesterol (TC), triglycerides (TG), low-density lipoprotein (LDL), very low-density lipoprotein (VLDL) and a significant increase in high-density lipoprotein (HDL). It also resulted in the improvement of liver oxidative biomarkers and decreased the production of inflammatory markers (TNF-$\alpha$, IFN-$\gamma$, IL-10 and IL-1$\beta$). These effects might be beneficial to further extend the application of wheat bran as a potential prebiotic or dietary supplement for hyperlipidemia treatment. The mechanism of *Spirulina maxima* extracts and Lactobacillus fermented wheat bran extract on serum lipids, inflammatory markers and liver enzymes needs further investigation.

**Author Contributions:** A.A. conceptualization, methodology, tables, figures, writing, review and editing; M.F.E. methodology, writing; S.I. analyzing data and reviewing; E.S.A. resources, software, validation. All authors have read and agreed to the published version of the manuscript.

**Funding:** The research received no external funding.

**Institutional Review Board Statement:** Not applicable.

**Informed Consent Statement:** Not applicable.

**Data Availability Statement:** Available on request.

**Conflicts of Interest:** The authors declare no conflict of interest.

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
