# Peer review of "Hypolipidemic, Antioxidant and Immunomodulatory Effects of Lactobacillus casei ATCC 7469-Fermented Wheat Bran and Spirulina maxima in Rats Fed a High-Fat Diet"

_fermentation, doi:10.3390/fermentation8110610_

Round 1

Reviewer 1 Report

This article is scientifically interesting and valuable. However, there are several points need to be clarified or revised. The manuscript could be reconsidered for publication after a major revision and clarification of the points given as follows: 

1.  Author used Sprague–Dawley rats to evaluate the effect Spirulina maxima extract and Lactobacillus casei fermented wheat bran extract in hyperlipidemia. However, hamster is recognized as the common animal model for investigating hypolipidemic effect. Could author illustrate to use Sprague–Dawley rats instead of hamster?

2.   In this study, Spirulina maxima extract and Lactobacillus casei fermented wheat bran extract may ameliorate hyperlipidemia. The functional compounds in the extracts and the mechanism of hypolipidemic effect should be analyzed to improve the quality of this article which could be reconsidered for publication.

3.  Figure 1 and Figure 2 are not clear that should be revised carefully for publication.

Author Response

Comments and Suggestions for Authors

This article is scientifically interesting and valuable. However, there are several points need to be clarified or revised. The manuscript could be reconsidered for publication after a major revision and clarification of the points given as follows: 

  1. Author used Sprague–Dawley rats to evaluate the effect Spirulina maximaextract and Lactobacillus casei fermented wheat bran extract in hyperlipidemia. However, hamster is recognized as the common animal model for investigating hypolipidemic effect. Could author illustrate to use Sprague–Dawley rats instead of hamster?

Thanks for reviewer’s valuable comments

Sprague–Dawley rats were available in our lab, in our future studies, we will do your recommendation and use hamsters

Also, there are many research articles which use Sprague–Dawley rats for investigating hypolipidemic effect of

………….

.000.0Lactobacillus and Bifidobacteria such as:

1) Xie N., Cui Y., Yin Y.N., Zhao X., Yang J.W., Wang Z.G., Fu N., Tang Y., Wang X.H., Liu X.W., Wang C.L., Lu F.G. Effects of two Lactobacillus strains on lipid metabolism and intestinal microflora in rats fed a high-cholesterol diet. BMC Complementary and Alternative Medicine 2011, 11:53

2) Salaj R., Štofilová J., Šoltesová A., Hertelyová Z., Hijová E., Bertková I., Strojný L.,  KruDliak P., Bomba A.. The Effects of Two Lactobacillus plantarum Strains on Rat Lipid Metabolism Receiving a High Fat Diet. The Scientific World Journal Volume 2013, Article ID 135142, 7 pages http://dx.doi.org/10.1155/2013/135142

3) An H.M., Park S.Y. , Lee D.K. , Kim J.R. , Cha M.K. , Lee  S.W. , Lim H.T., Kim K.J., Ha N.J.. Antiobesity and lipid-lowering effects of Bifidobacterium spp. in high fat diet-induced obese rats. Lipids in Health and Disease 2011, 10:116

4) Y. M. Choi1 , S. H. Bae2 , D. H. Kang3 and H. J. Suh4 Hypolipidemic Effect of Lactobacillus Ferment as a Functional Food Supplement. Phytother. Res. 2006,20, 1056–1060

  1. In this study, Spirulina maximaextract and Lactobacillus casei fermented wheat bran extract may ameliorate hyperlipidemia. The functional compounds in the extracts and the mechanism of hypolipidemic effect should be analyzed to improve the quality of this article which could be reconsidered for publication.

Thanks for reviewer’s valuable comments

More discussion about the functional compounds in Spirulina maxima extract and Lactobacillus casei fermented wheat bran extract and the possible mechanism of hypolipidemic effect was added to the revised manuscript and highlighted in the text with yellow

Further experimental analysis of extract’s components and possible mechanisms will be done in our next research.

  1. Figure 1 and Figure 2 are not clear that should be revised carefully for publication.

Thanks for reviewer’s valuable comments

Figure 1 and Figure 2 were improved in the revised manuscript.

Reviewer 2 Report

The paper entitled "Hypolipidemic, antioxidant and immunomodulatory effects of 2 Lactobacillus casei ATCC 7469 fermented wheat bran and Spir-3 ulina maxima in rats fed on high cholesterol diet" altough interesting needs to be deeply reviewed and improved. Here are some points:

1.The paper needs a deep english gramatical revision

2. Use "adverse side effects"

3. The mice were randomly housed in polypropylene cages at a temperature of 22±2ËšC, 93 and a light period of 12:00 to 12:00 and permitted to acclimatize for ten days before the 94 experiment. During the 10-day acclimation period, the animals were fed diet ad libitum 95 daily. After that the mice were randomized into six groups(n = 5/group) treated daily for 96 seven weeks as follows. Are the authors working in mice.

4. Use scale barrs in figure 3

5. Be consistent in the use of "Lactobacillus"

6. Discussion as well as conclusion could be improved

Author Response

Comments and Suggestions for Authors

The paper entitled "Hypolipidemic, antioxidant and immunomodulatory effects of 2 Lactobacillus casei ATCC 7469 fermented wheat bran and Spir-3 ulina maxima in rats fed on high cholesterol diet" altough interesting needs to be deeply reviewed and improved. Here are some points:

1.The paper needs a deep english gramatical revision

Thanks for reviewer’s valuable comments

English language was revised using AJE Grammar Check and by native English speaker (Dr. Sibel Irmak,one of co-authors)..

  1. Use "adverse side effects"

Thanks for reviewer’s valuable comments

It was corrected in the revised manuscript.

  1. The mice were randomly housed in polypropylene cages at a temperature of 22±2ËšC, 93 and a light period of 12:00 to 12:00 and permitted to acclimatize for ten days before the 94 experiment. During the 10-day acclimation period, the animals were fed diet ad libitum 95 daily. After that the mice were randomized into six groups(n = 5/group) treated daily for 96 seven weeks as follows. Are the authors working in mice.

Thanks for reviewer’s valuable comments

The experiment was done on rats. Sorry for that unintended mistake, Mice was replaced by rats in the revised manuscript.

  1. Use scale bars in figure 3

Thanks for reviewer’s valuable comments

Scale bars was used in Figure 3

  1. Be consistent in the use of "Lactobacillus"

Thanks for reviewer’s valuable comments

It was checked in the revised manuscript

  1. Discussion as well as conclusion could be improved

Thanks for reviewer’s valuable comments

Discussion and conclusion were improved and highlighted in the text with yellow

Reviewer 3 Report

The topic is interesting and the research work is globally well designed and performed. However, the manuscript is full of typos, repetition, misuse of brackets and abbreviations: this fact make the paper not scientific and difficult to evaluate. I recommend to the authors a massive review before resubmitting it. 

Round 2

Reviewer 1 Report

No  comments. 

Reviewer 2 Report

The paper has been significantly imporved and it is orginial and easy to follow 

Reviewer 3 Report

All the reviewers' proposals were taken into consideration, significantly improving the manuscript. For this reason I believe that now the paper is ready to be accepted for publication.